# Chitosan Microparticles Enhance the Intestinal Release and Immune Response of an Immune Stimulant Peptide in *Oncorhynchus mykiss*

**DOI:** 10.3390/ijms241914685

**Published:** 2023-09-28

**Authors:** Iván González-Chavarría, Francisco J. Roa, Felipe Sandoval, Carolina Muñoz-Flores, Tomas Kappes, Jannel Acosta, Romina Bertinat, Claudia Altamirano, Ariel Valenzuela, Oliberto Sánchez, Katherina Fernández, Jorge R. Toledo

**Affiliations:** 1Biotechnology and Biopharmaceuticals Laboratory, Departamento de Fisiopatología, Facultad de Ciencias Biológicas, Universidad de Concepción, Víctor Lamas 1290, Concepción 4030000, Chile; ivangonzalez@udec.cl (I.G.-C.); fraroa@udec.cl (F.J.R.); felisandoval@udec.cl (F.S.); carolinaamunoz@udec.cl (C.M.-F.); janacosta@udec.cl (J.A.); romibert@gmail.com (R.B.); osanchez@udec.cl (O.S.); 2Laboratory of Biomaterials, Departamento de Ingeniería Química, Facultad de Ingeniería, Universidad de Concepción, Barrio Universitario s/n, Concepción 4030000, Chile; tkappes@udec.cl (T.K.); kfernandeze@udec.cl (K.F.); 3Laboratorio de Cultivos Celulares, Escuela de Ingeniería Bioquímica, Pontificia Universidad Católica de Valparaíso, Valparaíso 2362803, Chile; claudia.altamirano@pucv.cl; 4Laboratory of Fish Culture and Aquatic Pathology, Department of Oceanography, Faculty of Natural and Oceanographic Sciences, Universidad de Concepción, Víctor Lamas 1290, Concepción 4030000, Chile; avalenz@udec.cl

**Keywords:** chitosan microparticles, immunostimulant peptides, rainbow trout, antibiotic resistance

## Abstract

The aquaculture industry is constantly increasing its fish production to provide enough products to maintain fish consumption worldwide. However, the increased production generates susceptibility to infectious diseases that cause losses of millions of dollars to the industry. Conventional treatments are based on antibiotics and antivirals to reduce the incidence of pathogens, but they have disadvantages, such as antibiotic resistance generation, antibiotic residues in fish, and environmental damage. Instead, functional foods with active compounds, especially antimicrobial peptides that allow the generation of prophylaxis against infections, provide an interesting alternative, but protection against gastric degradation is challenging. In this study, we evaluated a new immunomodulatory recombinant peptide, CATH–FLA, which is encapsulated in chitosan microparticles to avoid gastric degradation. The microparticles were prepared using a spray drying method. The peptide release from the microparticles was evaluated at gastric and intestinal pH, both in vitro and in vivo. Finally, the biological activity of the formulation was evaluated by measuring the expression of *il-1β*, *il-8*, *ifn-γ*, *Ifn-α*, and *mx1* in the head kidney and intestinal tissues of rainbow trout (*Oncorhynchus mykiss*). The results showed that the chitosan microparticles protect the CATH–FLA recombinant peptide from gastric degradation, allowing its release in the intestinal portion of rainbow trout. The microparticle-protected CATH–FLA recombinant peptide increased the expression of *il-1β*, *il-8*, *ifn-γ*, *ifn-α*, and *mx1* in the head kidney and intestine and improved the antiprotease activity in rainbow trout. These results suggest that the chitosan microparticle/CATH–FLA recombinant peptide could be a potential prophylactic alternative to conventional antibiotics for the treatment of infectious diseases in aquaculture.

## 1. Introduction

The aquaculture industry continuously increases fish production to provide enough product to maintain fish consumption worldwide [1]. Still, the increased production generates susceptibility to infectious diseases that cause losses of millions of dollars to the industry [2], risking its stability and production feasibility [3]. Conventional treatments are based on antibiotics and antivirals to reduce the incidence of pathogens, but they have disadvantages, such as antibiotic resistance generation, antibiotic residues in fish, and environmental damage [4,5,6,7,8]. Thus, searching for new effective therapeutic alternatives and, more importantly, developing preventative actions are permanent challenges in this field [9]. Accordingly, functional foods with active components that allow the generation of prophylaxis to prevent infections are an essential tool in the food industry [10,11]. For instance, the salmon industry has incorporated phytoactive elements [12] or special diets to improve the condition of fish against infections by viruses, bacteria, or parasites [13,14]. Immunostimulant compounds that increase the immune response in fish are a new alternative for generating functional food for fish [15].

Antimicrobial peptides have been considered an important source of different peptides with immune-stimulating, antiviral, bactericidal, and antiparasitic properties [16,17]. However, although antimicrobial peptides are an exciting alternative for generating functional foods, their protection against gastric degradation is a relevant challenge for their actual application in the industry [18,19]. Our research group recently developed a chimeric peptide composed of the cathelicidin and flagellin domains (CATH–FLA), which induces the expression of immunostimulatory molecules in the primary cultures of head kidney cells from rainbow trout. Interestingly, the design of the CATH–FLA allows easy expression and purification, unlike other antimicrobial peptides with immunostimulatory functions [20]. However, the use of this peptide in fish food requires protection against gastric degradation to maintain its bioactivity in vivo.

Encapsulating bioactive peptides in different materials can help overcome these challenges [21]. Studies have shown that encapsulation of bioactive peptides maintains their bioactivity and improves their stability [22,23,24,25,26,27,28,29,30,31]. One of the alternatives to peptide microencapsulation is the use of chitosan microparticles, which have been used as carriers for delivering drugs, proteins, and other bioactive agents to specific sites in the body of different species [32]. Chitosan is a natural biopolymer derived from chitin, a polysaccharide found in the shells of crustaceans, such as shrimp and crabs. Chitosan microparticles have unique properties that make them suitable for drug delivery applications [32,33]. They are biocompatible and biodegradable and have a high surface area-to-volume ratio, allowing efficient drug loading and release. Chitosan microparticles can also be modified to target specific cells or tissues in the body, making them an attractive option for targeted drug delivery [34]. Preparing chitosan microparticles typically involves using various techniques, such as emulsification, solvent evaporation, and spray drying [35]. These techniques allow for the formation of microparticles with controlled size, shape, and drug-release properties [36]. In drug delivery applications, chitosan microparticles can be loaded with a wide range of drugs, including anticancer agents, antibiotics, and anti-inflammatory agents. The use of chitosan microparticles as drug carriers has been shown to improve drug efficacy and reduce side effects as compared to conventional drug delivery methods. Chitosan microparticles have been evaluated as carriers for genetic material, and their application has been deeply analyzed [37]. 

Here we demonstrate that encapsulation of a new recombinant peptide, i.e., CATH–FLA, in chitosan microparticles, improves its protection from gastric degradation and allows its intestinal release while maintaining its bioactive properties in rainbow trout in vivo.

## 2. Results

### 2.1. Microparticle Generation

The morphological analysis using scanning electron microscope (SEM) showed microparticles with a pseudo-spherical morphology and varying roughness of their surfaces. The mean particle diameter (D50) determined using laser diffraction spectrometry was 11.98 ± 4.2 μm (N  =  2) (Figure 1), with 90% of the particles measuring less than 18.53  ±  3.2 μm (D90) and 10% measuring less than 7.56  ±  1.4 μm (D10). The production yield of the recovered particles was 65.9% ± 3.2%.

Fourier transform infrared spectroscopy-attenuated total reflectance (FTIR–ATR) infrared spectroscopy analysis was performed to analyze the presence of functional groups in the chitosan and the microparticles formulated after functionalization between the protein and the polymer. The FTIR spectra for loaded (blue) and empty (red) microparticles (Figure 2A) showed a peak at 1070 cm^−1^, corresponding to tension vibrations of the epoxy groups (C-O-C). A peak at ~1546 cm^−1^ referred to vibrations of the -NH2 groups present in the polymer structure. At 1646 cm^−1^, there was a peak attributed to H-O-H bending and a peak at 2880 cm^−1^ was associated with -CH stretching vibrations. Amino and hydroxyl groups at wavelengths between 3450 and 3100 cm^−1^ were present in these structures (represented by the yellow band). The comparison between both spectra showed a greater absorbance for the loaded microparticle than the empty microparticle (area defined in yellow), which suggests an interaction through hydrogen bonds between the chitosan and the peptides present in the microparticles (binding, not covalent). 

Thermogravimetric analysis (TGA) is a complementary technique used to reveal the composition and changes in the thermal stability of formulated materials. The thermograms corresponding to the formulated loaded and empty microparticles showed that, for both materials, the total weight loss was up to ~76% (Figure 2B). In the first stage of thermal decomposition, at a temperature below 100 °C, there was a weight loss of up to ~13% associated with the evaporation of the water absorbed in the sample and its stabilization. Subsequently, the loaded microparticle showed a weight loss of 36% at 165 °C, and the empty microparticle showed a weight loss of 30.3% at 170 °C, corresponding to the deacetylation and depolymerization of polymer chains [32,33], which was more noticeable in the empty microparticle due to the lack of the peptide. The total weight loss for both microparticles was 46%, which was associated with chitosan’s thermal and oxidative decomposition, coinciding with previous reports [33]. In both samples, the total weight loss was similar, meaning that the loaded microparticle had a thermal degradation like chitosan, and its structure was unaffected.

### 2.2. In Vitro Release of Chitosan/Peptide–BSA in Gastric and Intestinal pH

To determine the release of the peptide of interest from chitosan microparticles at gastric and intestinal pH (2 and 8, respectively), microparticles were co-loaded with bovine serum albumin (BSA) and the CATH–FLA (CATH–FLA–BSA) and incubated at pH 2 or pH 8 and in water, pH 7, for 0, 0.5, 1, 2, 6, and 24 h. Thus, the release of the CATH–FLA was analyzed indirectly through the detection of BSA using SDS-PAGE. The concentration of released BSA was calculated by interpolation on the BSA standard curve (Figure 3B). Data showed that the largest release of the CATH–FLA–BSA occurred mainly at pH 8 from 0 to 24 h, whereas a significantly lower release occurred at pH 2 and pH 7, particularly after 6 to 24 h of incubation (Figure 3A). These results suggest that most of the microparticle’s load is released at the intestinal pH and not at the gastric pH.

We also determined the release of the peptide from the microparticle in an in vitro experiment that more reliably represents the normal transit of food through the stomach and the intestine, as the formulation is intended to be administered orally to fishes. To this end, CATH–FLA–BSA-loaded microparticles were incubated first in buffer at pH 2 for 0, 0.25, 0.5, and 1 h, centrifuged, and then transferred to buffer at pH 8 for 0, 0.25, 0.5, 1, 6, and 24 h, centrifuged again and analyzed using SDS-PAGE. The results showed no significant changes in peptide–BSA release from the microparticles at pH 2 up to 1 h. In contrast, the release of peptide–BSA at pH 8 was significantly stimulated even at 0.25 h and up to 24 h, with the highest release at 0.5 h and a progressive decrease thereafter (Figure 4A,B). In conclusion, in vitro data suggest that our formulation passes gastric pH mostly unaltered and that the load is released at the intestinal pH.

### 2.3. In Vivo Analysis of CATH–FLA Peptide Release from Chitosan Microparticles

To determine whether the microparticles loaded with the immunostimulant peptide were protected from stomach acid degradation and then released in the intestinal portion of rainbow trout, we evaluated the release capacity of the formulation using infrared fluorescent dyes conjugated to BSA for the detection of BSA in vivo. We orally administered 100 μL (1 mg/mL) of the formulation, consisting of chitosan microparticles co-loaded with the CATH–FLA peptide and the fluorescent dye BSA–Remazol Blue, and the fishes were sacrificed 1 h later. Then, we analyzed the entire fish by placing it in the Pearl Impulse LI-COR infrared detection system to detect the Remazol Blue fluorescent dye. The results showed that the relative fluorescence of BSA–Remazol Blue increased 2.5-fold in the intestine close to the pyloric cecum and 1.8-fold in the distal intestine as compared to the control (CATH–FLA–BSA (without Remazol Blue microparticle) (Figure 5)). These results suggest that the immunostimulant peptide formulation is protected from degradation in the acidic environments of the stomach and the pyloric cecum and released in the intestinal portion of rainbow trout, supporting its potential use as an orally administered compound. 

To determine if the released immunostimulatory peptide produces a biological effect, the expression of interleukin *il*-*1β*, *il-8*, interferon (*ifn*)-γ, *ifn*-α, and myxoma resistance protein (*mx*) 1 were analyzed by means of qRT-PCR in the head kidney and intestinal tissues of rainbow trout treated for 6 or 12 days with chitosan microparticles loaded with the CATH–FLA or control. Briefly, the CATH–FLA peptide was expressed in BL21 bacteria and semi-purified, as previously described by our group [20]. As a control, the entire semi-purification process was carried out in extracts of BL21 bacteria without the induction of CATH–FLA expression. Both the CATH–FLA and the control extracts were microencapsulated as described above and used for the treatment.

Our results show that the microencapsulated immunostimulatory peptide significantly increased the expression of *il-1β*, *il-8*, *ifn-γ*, *ifn-α*, and *mx1* in the head kidney and intestinal tissues of rainbow trout (Figure 6). In parallel, although the control also shows an effect, this was of lower magnitude in most, if not all, of the transcripts analyzed (Figure 6), indicating the superiority of the CATH–FLA peptide in producing the intended effect.

In the intestinal tissue, there were significant increases of 3.6-fold, 3-fold, and 2.3-fold in the expression of *il-1β*, *il-8*, and *ifn-α*, respectively, as compared to the control group after 6 days of treatment. No significant changes were observed in the expression of *ifn-γ* and *mx1* at this time point. After 12 days of treatment, there were increases of 5.6-fold, 3.9-fold, 3.8-fold, 1.9-fold, and 1.7-fold in the expression of *il-1β*, *il-8*, *ifn-α*, *ifn-γ*, and *mx1*, respectively, as compared to the control group. 

In the head kidney tissue, there were significant increases of 2.2-fold, 7.7-fold, 5.5-fold, 2.8-fold, and 2.2-fold in the expression of *il-1β*, *il-8*, *ifn-α*, *ifn-γ*, and *mx1*, respectively, as compared to the control group at 6 days of treatment. After 12 days of treatment, there was an increase of 2.5-fold, 4.6-fold, 5.5-fold, and 8.6-fold in the expression of *il-1β*, *ifn-α*, *ifn-γ*, and *mx1*, respectively, as compared to the control group. No significant changes were observed in the expression of *il-8* in the head kidney tissue after 12 days of treatment. 

### 2.4. Antiprotease Activity in the Serum of Fishes Treated with the Formulation of the Microencapsulated Chitosan Immunostimulant Peptide 

Protease inhibitors in fish plasma are a vital defense against pathogens. These inhibitors, such as α1-antiprotease and α2-macroglobulin, block the action of proteolytic toxins produced by bacteria, preventing them from digesting fish tissue proteins. Therefore, we analyzed the antiprotease activity present in the serum from rainbow trout that were orally treated with empty (control) or CATH–FLA peptide-loaded chitosan microparticles after 6 and 12 days of treatment. The results showed a significant increase in antiprotease activity on day 12 of treatment as compared to the control group (Figure 7). These results suggest that the oral administration of the microencapsulated immunostimulant peptide increases the systemic antibacterial response in rainbow trout.

## 3. Discussion

Bioactive peptides are a promising area of research for the health, food, pharmaceutical, and cosmetic industries [38] as they offer various bioactivities, including antioxidant, antihypertensive, antimicrobial, anti-inflammatory, antiproliferative, and immunomodulatory activity [39]. However, their nutraceutical and commercial applications are limited by chemical degradation, interaction with food matrices, low water solubility, hygroscopicity, and palatability [21]. Encapsulation methods can be used to improve stability, protect the functionalities, and control the release of bioactive peptides [40]. Recently, our research group developed a chimeric peptide, the CATH–FLA peptide, which showed immunostimulatory activity on head kidney leukocytes from rainbow trout in vitro. In the present study, we moved one step forward and analyzed the in vivo effect of the CATH–FLA peptide. Given that gastric degradation of orally administered peptides is a critical problem, we also developed a chitosan microparticle to encapsulate the peptide for protection, supporting its use in fish [20]. 

Spray drying is a formulation approach that can be used to improve drug bioavailability, especially for those compounds with poor aqueous solubility and dissolution rate [41]. Furthermore, the bioactive compound can be effectively dried and formulated, improving its stability. Also, it could provide taste masking for drugs, an important area in functional food development in fish feed, because the palatability may affect food acceptance [42]. On the other hand, one of the main disadvantages of spray drying is the risk of denaturation and aggregation due to the high temperature and shear forces involved in the process [43]. However, our in vivo assay in *Oncorhynchus mykiss* demonstrated that the chitosan microparticles protected the CATH–FLA peptide, preserving its immunomodulatory characteristics. Another important point was the composition of the microparticles used in this study, as ionically cross-linked chitosan/tripolyphosphate particles have been extensively tested in biomedical applications, showing that, after preparation at acidic pH (4.43), exposure to a slightly alkaline pH (8.90) results in their virtually complete dissociation into free chitosan chains [44]. This agrees with our in vitro release assay showing minimal release of the CATH–FLA peptide at pH 2 and pH 7 but significantly increased release within seconds of exposure at pH 8. These data were further confirmed in vivo. Altogether, we conclude that one of the most relevant results of this study is that the spray drying approach for microencapsulation has no effect on the bioactivity of the CATH–FLA peptide and that the chitosan-based microparticles not only protect the peptide from gastric degradation but also allow its release at the intended place, i.e., the fish intestine. The conservation of the immunostimulant activity of the CATH–FLA peptide protected by the chitosan microparticle was tested in vivo in rainbow trout via oral administration with the formulation. 

We observed an increase in *il-1β*, *il-8*, *ifn-γ*, *ifn-α* and *mx1* in the head kidney and intestinal tissues of the rainbow trout, with a systemic response evidenced by the increase in antiprotease activity in the serum of fish treated for 12 days. Induction of *il-1β* and *il-8* expression was previously observed in the in vitro study of the chimeric CATH–FLA peptide [20]. This response is consistent with evidence that both cathelicidin and flagellin individually stimulate the expression of these pro-inflammatory cytokines [44,45,46]. On the other hand, in this new study where we administered CATH–FLA microencapsulated in chitosan microparticles in vivo, this pro-inflammatory response was maintained. Therefore, this evidence suggests that CATH–FLA induces the expression of the cytokines *il-1β* and *il-8*, which is not affected by its formulation or the oral route of administration. In addition, genes associated with the antiviral response are up-regulated: *ifn-α* (IFN type I) and *mx1*. While flagellin and cathelicidin have not been reported to generate an antiviral response, the microencapsulated chimera did induce an increase in the expression of both genes. This suggests the activation of signaling pathways that are different from those of the peptide components, leading to the activation of different innate immune responses. Finally, *ifn-γ* expression was analyzed and was significantly increased after administration of the microencapsulated chimeric peptide. In salmonids, IFN-γ has been shown to be involved in the regulation of the Th1-type immune response [47], and there is evidence that flagellin and cathelicidin can induce its expression [48,49]. Taken together, these results, therefore, suggest that CATH–FLA microencapsulated in chitosan microparticles could induce a pro-inflammatory, antiviral, and Th1 differentiation response in vivo, which could contribute to the activation of the immune response and the elimination of potential intracellular pathogens.

Finally, our results demonstrated that serum antiprotease activity increases 12 days post-treatment with the formulation of the CATH–FLA peptide in chitosan microparticles. In fish, it indicates an enhancement of the immune response. For example, in a study on *Cyprinus carpio* after immunization, serum antiprotease activity significantly increased as compared to the untreated control group [50]. Similarly, in rainbow trout, serum antiprotease activity was up-regulated in response to various stimuli, including bacterial and viral infections [51,52]. Thus, an increase in serum antiprotease activity in fish indicates an enhancement of the immune response and can be used as a marker for immunocompetence.

In conclusion, our study strongly suggests that the microencapsulation of the recently developed CATH–FLA peptide in chitosan microparticles protects it from degradation, allowing the immunostimulatory peptide to enhance the innate immune response after oral administration to rainbow trout. This microencapsulated immunostimulatory peptide could be used in food formulation and as an adjuvant in fish vaccination, providing a potential strategy for improving the resistance of fish to disease. 

## 4. Materials and Methods

### 4.1. Microencapsulation of Immunostimulant Peptide

The CATH–FLA peptide was expressed in BL21 bacteria and semi-purified, as previously described by our group [20]. As a control for in vivo immunostimulant assay, the entire semi-purification process was carried out, and the extracts of BL21 bacteria without induction of CATH–FLA expression were used to formulate control microparticles. 

The microencapsulated formulations of the immunostimulant recombinant peptide CATH–FLA and the control were prepared using spray drying technology. Low molecular weight chitosan (50–190 kDa, Sigma-Aldrich, St. Louis, MO, USA) was added to deionized water containing 0.5% (*v*/*v*) acetic acid (Sigma-Aldrich, St. Louis, MO, USA) under continuous stirring at 30 °C for 2 h at 800 rpm. A homogeneous chitosan solution was mixed with or without a suspension containing the peptide (1 mg/mL) or peptide plus BSA (1:1) (2 mg/mL). The peptide had a molecular size of 15 kDa, and its detection was complex, using electrophoresis systems. Thus, the bovine serum albumin (BSA) was incorporated to analyze the release of the microparticles in vitro. Furthermore, BSA conjugated to Remazol Brilliant Blue R (BSA–Remazol blue) was utilized for an in vivo release assay. BSA–Remazol Blue can be used as an infrared fluorescent dye. In this sense, the conjugation of BSA with this dye allows us to see its release from microparticles in vivo using infrared detection systems, such as the Licor-Pearl impulse equipment (absorption 680 nm emission 690 nm). 

The chitosan concentration in the mixture was 1% (*w*/*v*). The resultant suspension was fed into a B-290 mini spray dryer (BÜCHI Labortechnik AG, Flawil, Switzerland) with the following settings: feeding temperature at 20 ± 2 °C; inlet temperature at 100 ± 2 °C; outlet temperature at 55 ± 5 °C; air flow rate at 536 L/h; feed flow rate at 5 mL/min; aspiration vacuum at 60%; nozzle diameter at 0.7 mm. The microparticles were collected and stored at −20 °C, and the empty microparticles were used as the negative control.

### 4.2. Microparticle Characterization

Scanning electron microscopy (SEM) was used to analyze the morphology of the microparticles at 100X and 1000X magnifications. The dry powder of the microparticles was placed onto metal plates and sputter-coated with gold (Thermo Fisher Scientific, Waltham, MA, USA). The mean hydrodynamic diameter and size distribution of the microparticles suspended in water was measured with dynamic light scattering using a Microtrac S3500 (Microtrac Inc., Montgomeryville, PA, USA). The spectrum of the synthesized chitosan and microparticles was obtained with Fourier transform infrared spectroscopy (FTIR) using a Perkin Elmer UATR Two FTIR Spectrometer in the 4000–500 cm^−1^ wavenumber range with 40 accumulated scans. This technique was used to identify the functional groups present in these structures. Thermogravimetric analysis (TGA) was used to determine the thermal stability of the chitosan and the microparticles obtained using the spray drying technique. This analysis was developed using a Cahn-Versatherm thermogravimetric analyzer (Thermo Fisher Scientific, Waltham, MA, USA) with a sensitivity of 0.1 μg at a heating rate of 10 °C/min under a nitrogen atmosphere (100 mL/min). The temperature was scanned from room temperature to 600 °C.

### 4.3. Release Determination of Chitosan Peptide–BSA Formulation in Gastric and Intestinal pH

Three solutions were generated to emulate the gastrointestinal environment and used for the release assays: 50 mM Tris-HCl buffer pH 2, 100 mM glycine-HCl buffer pH 8, and distilled water pH 7. For the release, 60 mg of microencapsulated chitosan peptide/BSA (Sigma-Aldrich, St. Louis, MO, USA) formulation was resuspended in 30 mL of each condition. The samples were incubated at 18 °C and agitated at 150 rpm. Then, 2 mL of each condition were collected at 0, 0.5, 1, 2, 6, and 24 h. The samples were centrifuged at 1000 rpm for 10 min, and the supernatant was recovered in clean tubes. The presence of BSA and, indirectly, of the CATH–FLA peptide, was analyzed using SDS-PAGE and the LICOR-CLX detection system (LI-COR, Inc., Lincoln, NE, USA).

### 4.4. In Vivo Release Analysis of Chitosan Peptide–BSA Formulation in Rainbow Trout

The BSA and the peptide were stained with Remazol Blue (Sigma-Aldrich, St. Louis, MO, USA) to allow their detection with the LI-COR equipment’s infrared detector, Pearl Impulse. The protocol previously described for preparing microcapsules (spray dryer) was followed for this purpose, but CATH–FLA–BSA–Remazol Blue was incorporated. The microcapsules were resuspended in PBS (pH 7) at a concentration of 1 mg/mL, and then, 100 µL of suspension was administered orally to rainbow trout of 30–35 g (donated by the Chilean salmon company Salmones Antartica S.A.), which had not been fed for 1 day. After 60 min, the fish were sacrificed by anesthetic overdose, and the infrared fluorescence of CATH–FLA–BSA–Remazol Blue released in the pyloric caecum, proximal intestine, and distal intestine portions was analyzed. As a control, microparticles of chitosan peptide–BSA without the Remazol Blue conjugation were administered. The animals were documented using the LICOR-Pearl Impulse equipment at 680 nm.

### 4.5. Analysis of In Vivo Biological Activity of CATH–FLA Microparticles

To evaluate the biological activity of chitosan–CATH–FLA microparticles, the expression of crucial cytokines associated with innate immune response was analyzed in juvenile rainbow trout, *Oncorhynchus mykiss* (30–35 g) (30 fish per group). Each fish was treated orally (tubed) with 1 mg/mL per fish of chitosan–CATH–FLA microparticles (treated group) or 1 mg/mL of total protein microparticles in 100 μL of PBS (pH 7). Treatments were administered every three days for up to 12 days. On the 0, 6, and 12 days, 10 fish from the control and treated groups were selected and sacrificed to obtain intestinal tissue and head kidney and serum samples. The intestinal tissue and head kidney were deposited in TRIzol (1 g of tissue by 1 mL of TRIzol) and homogenized on ice for 30 s using homogenizer HG-15D (Witeg, Germany). The total RNA was extracted according to the protocol recommended by the supplier (Invitrogen, Thermo Fisher Scientific, Waltham, MA, USA). 

The expressions of interleukin 8 (*il-8*), interleukin 1 beta (*il-1β*), and interferon-gamma (*ifn-γ*), three modulatory cytokines of the innate immune response in salmonids, and the expressions of two cytokines of antiviral response, interferon-alpha (*ifn-α*) and interferon-induced dynamin-like GTPase (*mx1*), were evaluated using real-time PCR (Table 1). The real-time PCR was performed using the AriaMX instrument (Agilent Technologies, Santa Clara, CA, USA) and the Brilliant II SYBR Green qRT-PCR Master Mix1-Step PCR kit according to the protocol recommended by the supplier (Agilent Technologies, Santa Clara, CA, USA). The results were analyzed as 2^−ΔΔCT^ relative quantification. The comparative threshold cycle values were normalized for EF1-α mRNA.

### 4.6. Antiprotease Activity

Fish plasma contains protease inhibitors, mainly α1-antiprotease and α2-macroglobulin. Many infectious bacteria in aquaculture produce proteolytic toxins that digest fish tissue proteins; hence, protease inhibitors play an essential role in the defense against pathogens in fish. We analyzed the antiprotease activity of the trout serum from fishes treated with the formulation of immune stimulant peptides or controls after 6 and 12 days.

To determine the antiprotease activity, 20 μL of serum samples were incubated with an equal volume of 5 mg/mL trypsin standard solution (Sigma-Aldrich, St. Louis, MO, USA) at 22 °C for 10 min. Subsequently, 200 μL of 0.1 M phosphate buffer pH 7.0 and 250 μL of 2% (*w*/*v*) azocasein (Sigma-Aldrich, St. Louis, MO, USA) were added, and the mixture was incubated for 1 h at 22 °C. Then, 500 μL of 10% (*v*/*v*) TCA were added and incubated at 22 °C for 30 min. The mixture was centrifuged at 6000× *g* for 5 min. The supernatant was transferred to a 96-well plate (Corning™ Costar™, Kennebunk, ME, USA) containing 100 μL/well 1 N NaOH. The absorbance was determined at 450 nm in an ELISA reader (Tecan Group Ltd., Männedorf, Switzerland). As a control (100% trypsin activity), the sample was replaced with phosphate buffer, and for the negative control, the sample and trypsin were replaced with phosphate buffer. The percentage inhibition of trypsin activity in each sample was calculated by comparing the control sample with the control of 100% trypsin activity.

### 4.7. Statistical Analyses

Data are presented as means ± standard deviation (SD). Statistical analysis was performed using GraphPad Prism 7.0 software. Multiple comparisons were analyzed using one-way ANOVA with a Dunnett post-test, and a *p* value < 0.05 was considered statistically significant.

## 5. Patents

WO2020077478A1: Composition for stimulating the immune system in fish, which contains recombinant peptides; immunostimulant recombinant peptides; and nucleotide sequences encoding same. 

## Figures and Tables

**Figure 1 ijms-24-14685-f001:**
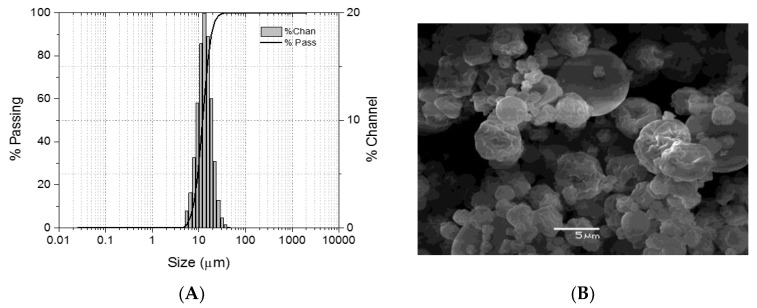
Characterization of loaded microparticles. Chitosan microparticles were loaded with the CATH–FLA peptide and analyzed using (**A**) laser diffraction spectrometry to determine size distribution and (**B**) scanning electron microscopy (SEM) to assess morphology.

**Figure 2 ijms-24-14685-f002:**
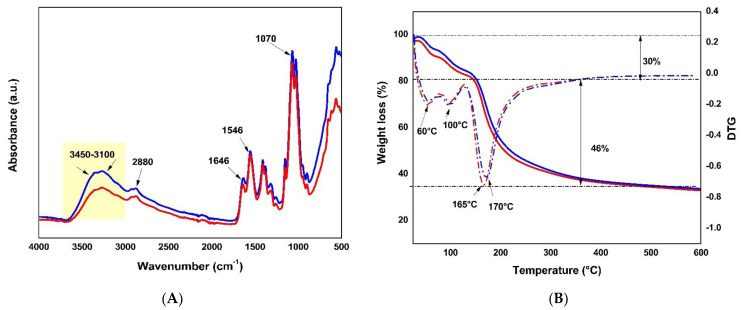
Analysis of the interaction between CATH–FLA and chitosan in microparticles. (**A**) Attenuated total reflectance–Fourier transform infrared (ATR–FTIR) spectroscopy and (**B**) thermogravimetric analysis were used to study peptide-loaded (blue) and empty (red) microparticles to determine thermogram thermogravimetry, TG (solid lines), and derivative thermogravimetry, DTG (dotted lines).

**Figure 3 ijms-24-14685-f003:**
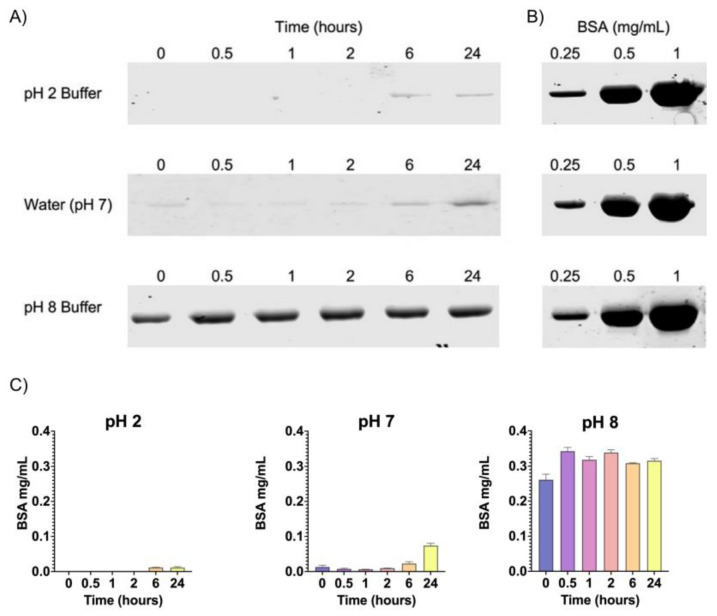
In vitro analysis of the release of CATH–FLA–BSA from chitosan microparticles. Chitosan microparticles were co-loaded with the CATH–FLA and bovine serum albumin (BSA) (CATH–FLA–BSA), and the release of the CATH–FLA was analyzed indirectly through the detection and semi-quantification of BSA (against BSA standards) using SDS-PAGE. (**A**) Loaded microparticles were incubated separately in buffer at gastric pH (2) or intestinal pH (8) and in water (pH 7) for 0, 0.5, 1, 2, 6, and 24 h, centrifuged, and the pellet was analyzed using SDS-PAGE (**B**) Standard curve of BSA for release quantification. (**C**) The graphs represent the concentration of BSA released at different incubation times in buffer at pH 2, pH 7 and pH 8. The data represent the means ± S.D. of three independent experiments.

**Figure 4 ijms-24-14685-f004:**
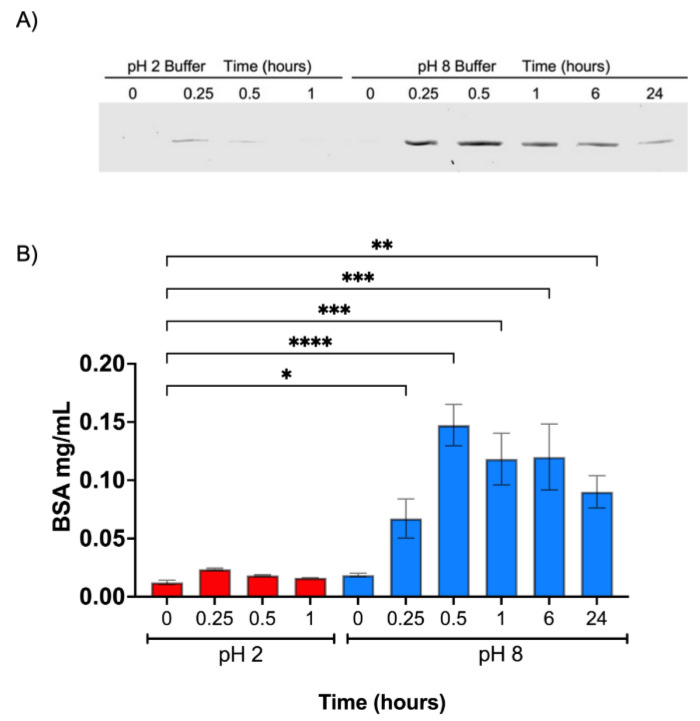
Temporal in vitro release. The chitosan microparticles were co-formulated with bovine serum albumin BSA and the CATH–FLA peptide (peptide–BSA). The same sample was deposited in pH 2 for 1 h and then exposed to pH 8. (**A**) The release was determined with SDS-PAGE 10% using BSA standards and the interpolation of BSA band intensity. (**B**) The graph showed the concentration of BSA released at different incubation times in buffer at pH 2 (in red) and pH 8 (in blue). The data represent the means ± S.D. of three independent experiments analyzed using one-way analysis of variance and Dunnett’s post-test (**** *p* ≤ 0.0001, *** *p* ≤ 0.001, ** *p* ≤ 0.01, * *p* ≤ 0.05).

**Figure 5 ijms-24-14685-f005:**
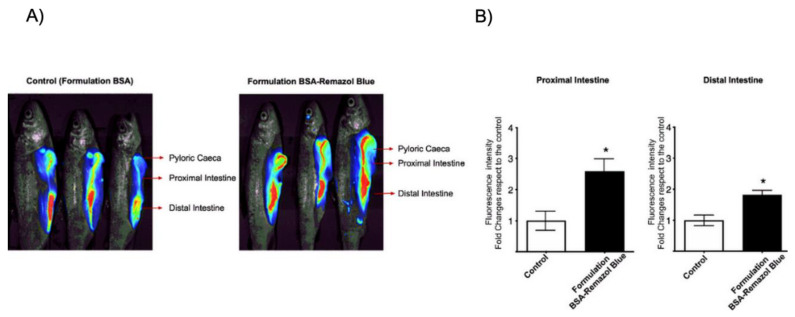
In vivo analysis of the release of CATH–FLA–BSA Remazol Blue from chitosan microparticles. Chitosan microparticles were co-loaded with the CATH–FLA and BSA without Remazol Blue (control) or the CATH–FLA–BSA conjugated to fluorescent dye Remazol Blue (CATH–FLA–BSA-Remazol Blue) and orally administered to rainbow trout. The release of the CATH–FLA–BSA–Remazol Blue was analyzed indirectly through the NIR fluorescence of Remazol Blue using the LI-COR infrared detection system. (**A**) Infrared fluorescence images show a gradient of Remazol Blue fluorescence of high (red), medium (green), and low (blue) intensity associated with the higher-to-lower release of the particle load, respectively, at different sites of the digestive system, pyloric caeca, proximal intestine, and distal intestine. (**B**) The highest fluorescence intensity of Remazol Blue (red areas) in the proximal and distal intestine was observed and semi-quantified. The mean value was assigned value 1 in control fishes (*n* = 3, white bars) and compared to the experimental fishes (*n* = 3, black bars). The experiment was repeated 3 times (* = *p* < 0.05).

**Figure 6 ijms-24-14685-f006:**
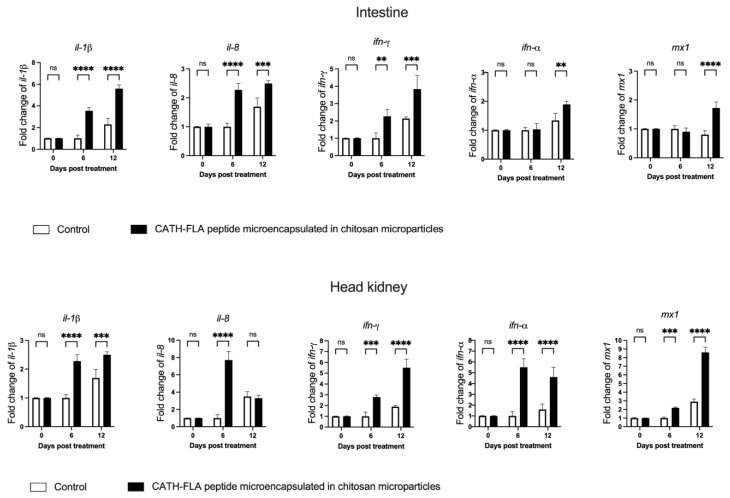
In vivo biological activity of the CATH–FLA microencapsulated in chitosan microparticles. Chitosan microparticles were loaded with the CATH–FLA or control (semi-purification process was carried out in extracts of BL21 bacteria without induction of CATH–FLA expression and the extract was microencapsulated and used as control). The microparticles were orally administered to rainbow trout every 3 days. After 6 and 12 days of treatment, fish were sacrificed, and the transcript levels of *il-1β*, *il-8*, *ifn-α*, *ifn-γ*, *and mx1* were determined in the head kidney and intestinal tissues. The relative expression of the selected markers was determined with qPCR using the 2^−ΔΔCT^ method, normalizing against elongation factor 1-α (*ef1-α*). Data represent the means ± S.D. of three independent experiments analyzed using one-way analysis of variance and Dunnett’s post-test (**** *p* ≤ 0.0001, *** *p* ≤ 0.001, ** *p* ≤ 0.01, ns = not significant).

**Figure 7 ijms-24-14685-f007:**
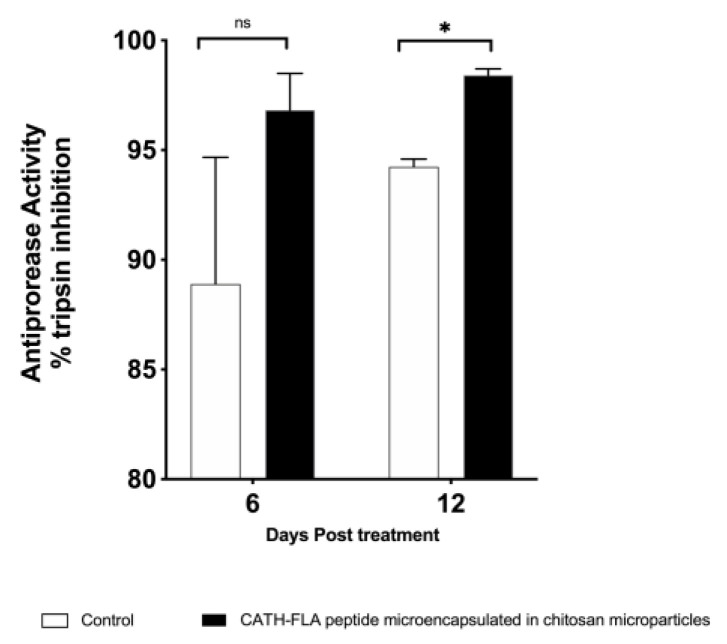
Antiprotease activity in serum from fishes treated with chitosan microparticles loaded with the immunostimulant peptide CATH–FLA or control microparticles. Antiprotease activity was measured in serum samples after 6 and 12 days of treatment. Data represent the means ± S.D. of three independent experiments analyzed using one-way analysis of variance and Dunnett’s post-test (* *p* ≤ 0.05, ns = not significant).

**Table 1 ijms-24-14685-t001:** Primers [48].

Gene	Forward (5′→ 3′)	Reverse (5′→3′)
*ifn-α*	TGGGAGGAGATATCACAAAGC	TCCCAGGTGACAGATTTCAT
*il-1β*	GCTGGAGAGTGCTGTGGAAGA	TGCTTCCCTCCTGCTCGTAG
*il-8*	ATTGAGACAGAAAGCAGACG	CGCTGACATCCAGACAAATCT
*ifn-γ*	CCGTACACCGATTGAGGACT	GCGGCATTACTCCATCCTAA
*mx1*	TGCCTCGTCTAGAAGAGCA	CTTTCCAGCTCGGCATGTG
*ef1-α*	GTGACACCGAAACTAAGCGAC	TGTAGATCAGATGGCCGGTG

## Data Availability

Not applicable, work under patent.

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
