# Peer review of "Chitosan Microparticles Enhance the Intestinal Release and Immune Response of an Immune Stimulant Peptide in Oncorhynchus mykiss"

_ijms, 2023, doi:10.3390/ijms241914685_

Round 1

Reviewer 1 Report

The manuscript by González-Chavarría and colleagues title Chitosan microparticles enhance the intestinal release and immune response of an immune stimulant peptide in Oncorhychus mykiss” is about the evaluation of a new immunomodulatory recombinant peptide CATH-FLA encapsulated in chitosan microparticles and administered by food. The results obtained indicate that chitosan microparticles protects the peptide from degradation of gastric and intestinal pH. Finally, the biological activity of the peptide CATH-FLA was preserved with the chitosan microparticles encapsulation.

In my opinion this work is interesting but does not imply much novelty and important contribution to the field, because there is a previous publications where the same authors evaluate the activity and the activation of immune response of the peptide CATH-FLA (doi:10.1016/j.fsi.2022.04.034) in fish. It is clear that in this manuscript the novelty is the evaluation of the oral administration in fish by chitosan microparticles and they saw the same result that in the paper mention before. They performed many experiments to determine the stability, protection and release of the peptide encapsulated by chitosan microparticles. However, there are many works where these properties of chitosan microparticles are published.

In addition, I found this manuscript difficult to read and understand because it is not well written (I recommend English revision), many information is missing in material and methods, I would add a table of genes with their sequences and accession number or reference, the discussion needs a revision of its content and better discussed and there are many mistakes such as:

Minnor mistakes:

-Define acronyms: SEM (Line 89), FTIR-ATR (line 96), all the genes in line 167 and 168

-Eliminate ., (line 91)

-Change  cm-1 as super-index such as cm-1 from line 99 to 103

-analysis is repeated two times in line 108, delate one

-in vitro and in vivo should be in italics please change them all for the whole manuscript.

- I believe that realise means release please change in line 140 and revised the whole manuscript and change them all.

- Genes should be written in lower case and italic in line 167 and 168 and change in the whole manuscript

-Line 192 per os administration ????

- the capitals are put in nonsense  for example line217 Spay for spray, Basic line 233, RPM line 291…

Figures:

-Some legends have a title (Figure 4, 5 and 6) and some have not (Figure 1,2 and 3). I suggest to add titles in the figures 1,2 and 3.

Legends are not well explained: Figure 1, add the magnification of the image; Figure2, indicate the meaning of dashed line and solid line in figure 2B and define DTG acronym.

Figure 3 is a complete disaster.  There are 2 A (A and a) that are not explained in the legend. In addition, this figure is so confusing because there are many figures that I can't identify their explanation with the legend and I don't know what they are. I recommend adding more letters and explaining properly each graph or figure in the legend.

-Figure 4 does not mention the meaning of the 2 graphs on the right in the legend. I suggest to add letters and explain in the legend.

-Figure 5. I do not understand the relative expression indicated on the y-axis of the graphs. Relative to what exactly? It is not explained the meaning of “microencapsulated formulation of total protein from BL21 cells”. This information is missing in material and methods and also in the legend. Also factor 1-α is incomplete “Elongation factor 1-α “is the correct name (line 186). Please change  anterior kidney for head kidney.

Figure 6. Authors did not mention what does the control consist of.

Results.

 Result 2.3 “in vivo formulation release assay”. It is bad explained and confusing for example “we treated fish orally with 100microl of the formulation “line 150. Which formulation?

Line 154, authors said that they compared with the control (empty microparticle) but the control in the figure 4 is formulation with BSA?.

Material and methods

The commercial references are missing in the whole section   for example; BSA line 292, using electrophoresis and infrared detection systems line 294, metal plates and sputter-coated with gold, line 304, Cahn-Versatherm thermogravi metric analyzer line 312, 50 mM Tris-HCl buffer pH 2.0, 100 mM glycine-HCl buffer pH 318 8.0, and distilled water pH 7.0. line 318, LICOR-CLX equipment´s line 324, remazol blue line 326, trizol method line 345, AriaMX instrument Line 352, anesthetic line 332, real time PCR line 347….

There are many techniques that are not well explained, not detailed and not indicated how they are done:

For example, the electrophoresis, trizol method, RT-PCR, fish plasma, antiproteasa activity, PCR condition and PCR analysis method…

 There is no indication of fish size, not indication of where they are purchased from or a section with ethics and animal welfare in in vivo assays. In addition, some treatments are not well explained for example “50 fish per group). Fish were treated orally with 1 mg/mL line 340” but how? tubed or feed?   1 mg/Ml per fish?” Treatments were administered every three days for 12 days “line 342 which treatment and how?

Discussion

In general discussion needs a revision of its content and better discussed.

Moreover, the paragraph “IFN-γ homologs have been identified in rainbow trout, and their biological activities have been characterized IFN-γ is one of the key cytokines involved in Th1 immune responses, and its orthologues have been discovered to be functionally conserved in fish 261 [52]”. Line 259-262 It is too long I would divide in two sentences.

Line 264 “unique member, have been identified in rainbow trout”. Which one?

Line 272” In fish, it indicates an enhancement of the immune response”. Explain better.

it is not well written ,I recommend English revision.

Author Response

We thank the reviewers for their observations and comments. These were received and are described and corrected in the new manuscript.

Best regards,

Reviewer 2 Report

 The manuscript "Chitosan microparticles enhance the intestinal release and immune response of an immune stimulant peptide in Oncorhynchus mykiss” prepared by Iván González-Chavarría and co-authors is a well-designed work aimed at creating a useful product for fish farming.

The introduction fully substantiates the need for approaches other than antibiotic therapy to prevent salmon infections and suggests a new approach to disease prevention - the use of biologically active peptides, in particular, immunostimulating compounds. The authors used their own chimeric peptide consisting of cathelicidin and flagellin domains (CATH-FLA), both of which have immunomodulatory properties, and this peptide appears to be a good choice. Chitosan-based microcapsules were used to deliver the peptide.

The "Results" section consistently and clearly outlines experiments on the characterization of microcapsules and their protective functions, for which a set of adequate methods was used. The authors obtained the results necessary for the development of an active drug: the resistance of microcapsules loaded with the peptide to the acidic contents of the stomach, the dynamics of the release of the peptide, and, most importantly, showed the effect on the production of cytokines in trout, confirming its immunostimulating effect.

The positive impression of the manuscript is spoiled by the careless design of the figure captions. Thus, to designate blocks of figures, captions are used in capital letters, while in the figures - small lowercase letters. For example: A), and (a). The interpretation of the meaning (*) in the figures is given only in the caption to Figure 5: The data represent the means ± S.D. of three independent experiments analyzed by one-way analysis of variance and Dunnett's post-test (***p≤0.001, **p≤0.01, *p≤0.05). All figure captions must be corrected.

The "Materials and Methods" section is clearly written and gives a comprehensive idea of how the work was done.

In total, the manuscript looks interesting and will undoubtedly be of interest to a wide range of IJMS readers working both in the field of creating carriers for peptide preparations and in the field of fish farming.

Author Response

(The authors gave the same response as above.)
